# GROUP ROBUSTNESS VIA ADAPTIVE CLASS-SPECIFIC SCALING

## ABSTRACT

Group distributionally robust optimization, which aims to improve robust accuracies such as worst-group or unbiased accuracy, is one of the mainstream algorithms to mitigate spurious correlation and handle dataset bias. Existing approaches have apparently improved robust accuracy, but, in fact, these performance gains mainly come from trade-offs at the expense of average accuracy. To control the trade-off flexibly and efficiently, we first propose a simple class-specific scaling strategy, directly applicable to existing debiasing algorithms without additional training. We also develop an instance-wise adaptive scaling technique to overcome the trade-off and improve the performance even further in terms of both accuracies. Our approach reveals that a naïve ERM baseline matches or even outperforms the recent debiasing methods by only adopting the class-specific scaling technique. Then, we employ this technique to evaluate the performance of existing algorithms in a comprehensive manner by introducing a novel unified metric that summarizes the trade-off between the two accuracies as a scalar value. By considering the inherent trade-off and providing a performance landscape, our approach delivers meaningful insights into existing robust methods beyond the robust accuracy only. We perform experiments on the datasets in computer vision and natural language processing domains and verify the effectiveness of the proposed frameworks.

## 1 INTRODUCTION

Machine learning models exhibit remarkable performance in various tasks via empirical risk minimization (ERM). However, they often suffer from spurious correlations and dataset biases, resulting in poor performance in the classification of minority groups despite high average accuracies. For instance, because digits and foreground colors have a strong correlation in the colored MNIST dataset (Arjovsky et al., 2019; Bahng et al., 2020), a trained model learns unintended patterns in input images and performs poorly in classifying digits that belong to minority groups, in other words, when the colors of the digits are rare in the training dataset.

Since it is well-known that spurious correlation leads to poor generalization performance in minority groups, group distributionally robust optimization (Sagawa et al., 2020) has been widely used to mitigate the algorithmic bias. Numerous approaches (Huang et al., 2016; Sagawa et al., 2020; Seo et al., 2022a; Nam et al., 2020; Sohoni et al., 2020; Levy et al., 2020; Liu et al., 2021) have demonstrated high robust accuracies such as worst-group or unbiased accuracies in a variety of tasks and datasets, but, while they clearly sacrifice the average accuracy, there has been little exploration for a comprehensive evaluation that considers both metrics jointly. Figure 1 demonstrates the existing trade-offs of algorithms.

This paper addresses the limitations of the current research trends and starts with introducing a simple post-processing technique, *robust scaling*, which efficiently performs class-specific scaling on prediction scores and conveniently controls the trade-off between robust and average accuracies at test time. It allows us to identify any desired performance points, *e.g.*, ones in terms of average accuracy, unbiased accuracy, worst-group accuracy, or balanced accuracy, on the accuracy trade-off curve obtained from a pretrained model with negligible extra computational overhead. The proposed robust-scaling method can be easily plugged into various existing debiasing algorithms to improve the desired target objectives within the trade-off. One interesting observation is that, by adopting the proposed robust scaling, even the ERM baseline accomplishes competitive performance without

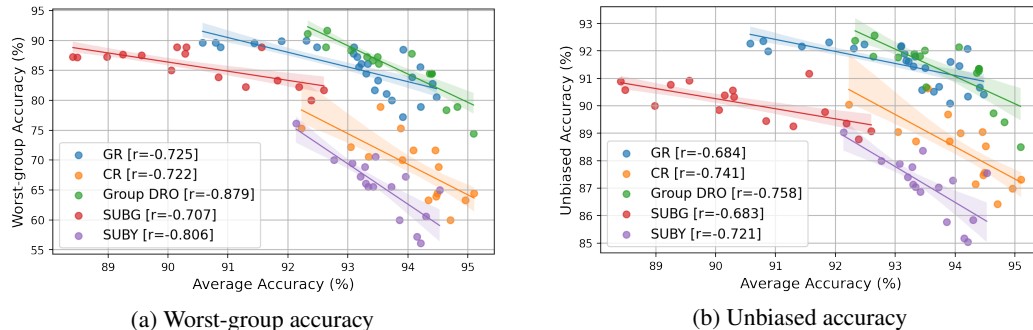

(a) Worst-group accuracy

(b) Unbiased accuracy

Figure 1: The scatter plots illustrate trade-offs between robust and average accuracies of existing algorithms with ResNet-18 on CelebA. We visualize the results from multiple runs of each algorithm and present the relationship between the two accuracies. The lines denote the linear regression results of individual algorithms and $r$ in the legend indicates the Pearson coefficient correlation.

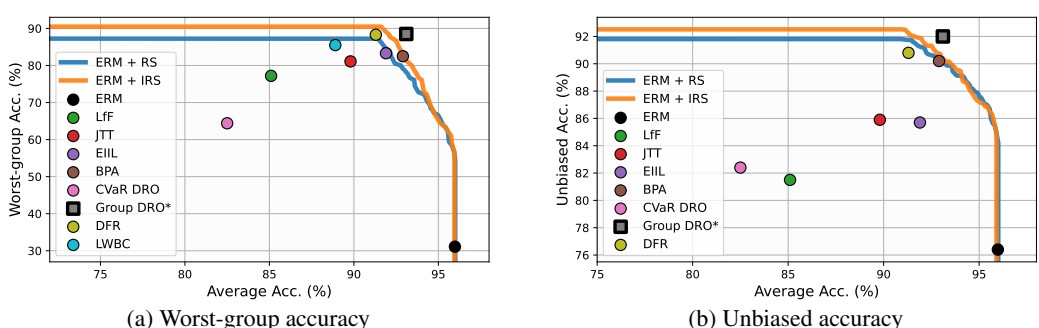

(a) Worst-group accuracy

(b) Unbiased accuracy

Figure 2: Comparison between the baseline ERM and existing debiasing approaches with ResNet-50 on CelebA. Existing works have improved robust accuracy substantially compared to ERM, but our robust scaling strategies such as RS and IRS enable ERM to catch up with or even outperform them without further training.

extra training compared to the recent group distributionally robust optimization approaches (Liu et al., 2021; Nam et al., 2020; Sagawa et al., 2020; Kim et al., 2022; Seo et al., 2022a; Creager et al., 2021; Levy et al., 2020; Kirichenko et al., 2022; Zhang et al., 2022), as illustrated in Figure 2. Furthermore, we propose a more advanced robust scaling algorithm applicable to each example adaptively based on its cluster membership at test time to maximize performance. Our instance-wise adaptive scaling strategy is effective to overcome the trade-off and achieve performance gains in terms of both accuracies.

By taking advantage of the robust scaling technique, we develop a novel comprehensive evaluation metric that consolidates the trade-off of the algorithms for group robustness, providing a unique perspective of group distributionally robust optimization. To this end, we first argue that comparing the robust accuracy without considering the average accuracy is incomplete and a unified evaluation of debiasing algorithms is required. For a comprehensive performance evaluation, we introduce a new measure referred to as *robust coverage*, which conveniently considers the trade-off between average and robust accuracies from the Pareto optimal perspective and summarizes the performance of each algorithm with a scalar value.

**Contribution** We present a simple but effective approach for group robustness by analyzing trade-off between robust and average accuracies. Our framework captures the full landscape of robust-average accuracy trade-offs, facilitates understanding the behavior of existing debiasing techniques, and provides a way for optimizing the arbitrary objectives along the trade-off without extra training. We emphasize that our framework does not solely focus on performance improvement in robust accuracy; more importantly, **our method reveals the trade-offs inherent in existing debiasing methods, facilitates to identify desired performance point depending on target objectives, and**

**paves a way for accurate, fair, and comprehensive evaluation of group robustness.** Our main contributions are summarized as follows.

- We propose a training-free class-specific scaling strategy to capture and control the trade-off between robust and average accuracy with negligible computational cost. This approach allows us to optimize a debiasing algorithm towards arbitrary objectives within the trade-off on top of any existing models.

- We develop an instance-wise robust scaling algorithm by extending the original class-specific scaling with joint consideration of feature clusters. This technique is effective to overcome the trade-off and improve both robust and average accuracy.

- We introduce a novel comprehensive and unified performance evaluation metric using the robust scaling method, which summarizes the trade-off as a scalar value from the Pareto optimal perspective.

- The extensive experiments analyze the characteristics of existing methods and validate the effectiveness of our frameworks on the multiple standard benchmarks.

## 2  RELATED WORKS

Mitigating spurious correlation has emerged as an important problem in many areas in machine learning. Sagawa et al. (2020) propose group distributionally robust optimization (Group DRO) with a practical assumption; training examples are given in groups and an arbitrary test distribution is represented by a mixture of these groups. Because models relying on a spurious correlation yield poor worst-group accuracy, Group DRO, which aims to maximize worst-group accuracy, is widely adopted to deal with the limitation and the methods can be mainly categorized into three folds.

**Sample reweighting**    The most popular approaches are assigning different training weights to each samples to focus on the minority groups, where the weights are based on the group frequency or loss. Group DRO (Sagawa et al., 2020) minimizes the worst-group loss by reweighting based on the average loss per group. Although Group DRO achieves robust results against group distribution shifts, it requires training examples with group supervision. To handle this limitation, several unsupervised approaches have been proposed that do not exploit group annotations. George (Sohoni et al., 2020) and BPA (Seo et al., 2022a) extend Group DRO in an unsupervised way, where they first train the ERM model and use this model to infer pseudo-groups via feature clustering. CVaR DRO (Levy et al., 2020) minimizes the worst loss over all $\alpha$-sized subpopulations, which upperbounds the worst-group loss over the unknown groups. LfF (Nam et al., 2020) simultaneously trains two models, one is with generalized cross-entropy and the other one is with standard cross-entropy loss, and reweights the examples based on their relative difficulty score. JTT (Liu et al., 2021) conducts a two-stage procedure, which upweights the examples that are misclassified by the first-stage model. Idrissi et al. (2022) analyze simple data subsampling and reweighting baselines based on group or class frequency to handle dataset imbalance issues. LWBC (Kim et al., 2022) employs an auxiliary module to identify bias-conflicted data and assigns large weights to them.

**Representation learning**    On the other hand, some approaches aim to learn debiased representations directly to avoid spurious correlation. ReBias (Bahng et al., 2020) adopts Hilbert-Schmidt independence criterion (Gretton et al., 2005) to learn feature representations independent of the pre-defined biased representations. Cobias (Seo et al., 2022b) defines conditional mutual information between feature representations and group labels as a bias measurement, and employ it as a debiasing regularizer. IRM (Arjovsky et al., 2019) learns invariant representations to diverse environments, where the environment variable can be interpreted as the group. While IRM requires the supervision of the environment variable, EIIL (Creager et al., 2021) and PGI (Ahmed et al., 2020) are the unsupervised counterparts, which assign each training example to the environment that violates the IRM objective.

**Post-processing**    While most previous studies are in-processing approaches that perform feature representation learning or sample reweighting during training to improve group robustness, our framework does not fall into those categories; it deals with group robust optimization by simple post-processing with class-specific score scaling, which does not require any additional training. Similar to our method, some post-processing techniques, including temperature (Guo et al., 2017) or Platt (Platt, 2000) scaling, are widely exploited in the literature of confidence calibration, but it

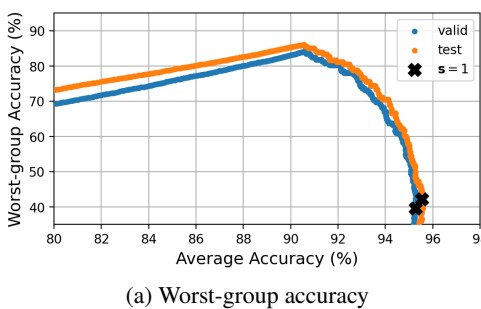 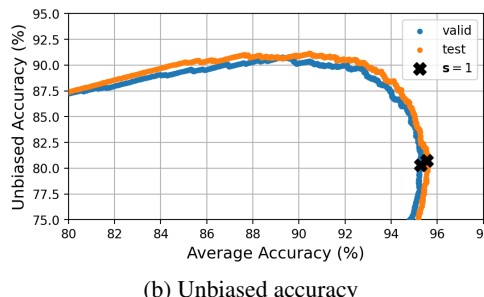

(a) Worst-group accuracy         (b) Unbiased accuracy

Figure 3: The relation between the robust and average accuracies obtained by varying the class-specific scaling factor **s** with ERM on CelebA. The black marker denotes the original point, where the uniform scaling is applied.

is not applicable in our task because it scales the prediction scores equally for each class and does not change the label predictions. Recently, several post-hoc approaches are proposed to retrain the last layer of the model with group-balanced dataset (Kirichenko et al., 2022) or correct the final logits (Liu et al., 2023), but they still require additional training different from our framework.

## 3 PROPOSED APPROACH

This section first presents a class-specific scaling technique, which captures the trade-off landscape and identifies the optimal performance points for desired objectives along the trade-off curve. We also propose an instance-wise class-specific scaling approach to overcome the trade-off and further improve the performance. Based on the scaling strategy, we introduce a novel and intuitive measure for evaluating the group robustness of an algorithm with consideration of the trade-off.

### 3.1 PROBLEM SETUP

Consider a triplet $(x, y, a)$ with an input feature $x \in \mathcal{X}$, a target label $y \in \mathcal{Y}$, and an attribute $a \in \mathcal{A}$. We construct a group based on a pair of a target label and an attribute, $g := (y, a) \in \mathcal{Y} \times \mathcal{A} =: \mathcal{G}$. We are given $n$ training examples without attribute annotations, *e.g.*, $\{(x_1, y_1), ..., (x_n, y_n)\}$, while $m$ validation examples have group annotations, , *e.g.*, $\{(x_1, y_1, a_1), ..., (x_m, y_m, a_m)\}$, for selecting scaling parameters. This assumption is known to be essential for model selection or hyperparameter tuning (Sagawa et al., 2020; Liu et al., 2021; Nam et al., 2020; Idrissi et al., 2022) although not desirable for the practicality of algorithms. However, we will show that our algorithm works well with a few examples with attribute annotations in the validation set; considering such marginal labeling cost, our approach is a meaningful step to deal with notorious bias problems in datasets and models.

Our goal is to learn a model $f_\theta(\cdot) : \mathcal{X} \to \mathcal{Y}$ that is robust to group distribution shifts. To measure the group robustness, we typically refer to the robust accuracy such as unbiased accuracy (UA) and worst-group accuracy (WA). The definitions of UA and WA require the group-wise accuracy (GA), which is formally given by

$$\text{GA}_{(r)} := \frac{\sum_i \mathbb{1}(f_\theta(\mathbf{x}_i) = y_i, g_i = r)}{\sum_i \mathbb{1}(g_i = r)}, \qquad (1)$$

where $\mathbb{1}(\cdot)$ denotes an indicator function and $\text{GA}_{(r)}$ is the accuracy of the $r^{\text{th}}$ group samples. Then, the robust accuracies are defined by

$$\text{UA} := \frac{1}{|\mathcal{G}|} \sum_{r \in \mathcal{G}} \text{GA}_{(r)} \quad \text{and} \quad \text{WA} := \min_{r \in \mathcal{G}} \text{GA}_{(r)}. \qquad (2)$$

The goal of the group robust optimization is to ensure robust performance in terms of UA or WA regardless of the group membership of a sample.

## 3.2 CLASS-SPECIFIC ROBUST SCALING

As shown in Figure 1, there exists a clear trade-off between robust and average accuracies for each algorithm. To understand its exact behavior, we design a simple non-uniform scaling method for the scores corresponding to individual classes. This strategy can change the final decision of the classifier; if we upweight the prediction scores of minority classes, the sample may be classified into one of the minority classes even with its low initial scores. Thus, the worst-group accuracy may increase at the expense of the average accuracy, resulting in a desirable trade-off for group robustness. Formally, the prediction with the class-specific scaling is given by

$$\arg\max_c \ (\mathbf{s} \odot \hat{\mathbf{y}})_c, \tag{3}$$

where $\hat{\mathbf{y}} \in \mathbb{R}^C$ is a prediction score vector over $C$ classes, $\mathbf{s} \in \mathbb{R}^C$ is a $C$-dimensional scaling coefficient vector, and $\odot$ denotes the element-wise product operator.

Based on the ERM model, we obtain a set of the average and robust accuracy pairs using a wide range of the class-specific scaling factors and illustrate their relations in Figure 3. The black markers indicate the point with a uniform scaling, $i.e.$, $\mathbf{s} = (1, \ldots, 1) \in \mathbb{R}^C$. The graphs show that a simple class-specific scaling effectively captures the landscape of the trade-off between the two accuracies. This validates the ability to identify the desired Pareto optimal points between the robust and average accuracies in the test set by following two simple steps: 1) finding the optimal class-specific scaling factors that maximize the target objective (UA, WA, or AA) in the validation set, and 2) apply the scaling factors to the test set[1]. We refer to this scaling strategy for robust prediction as *robust scaling*.

To identify the optimal scaling factor $\mathbf{s}$, we perform a greedy search, where we first identify the best scaling factor for a class and then determine the optimal factors of the remaining ones sequentially conditioned on the previously estimated scaling factors. The greedy search is sufficient for finding good scaling factors partly because there are many different near-optimal solutions. Thanks to the simplicity of the process, the entire procedure takes negligible time even in large-scale datasets with multiple classes. Note that the robust scaling is a post-processing method, so it can be easily applied to existing robust optimization techniques without extra training; our method can find any desired performance points on the trade-off envelop given a trained model. For example, even when dealing with multiple tasks, our robust scaling approach is flexible enough to handle the situation; we only need to apply a scaling factor optimized for each target objective, leaving the trained model unchanged. Meanwhile, existing robust optimization methods have limited flexibility and require to training separate models for each target objective.

## 3.3 INSTANCE-WISE ROBUST SCALING

The optimal scaling factor can be applied adaptively to each test example and the instance-specific scaling has the potential to overcome the trade-off and improve accuracy even further. Previous approaches (Seo et al., 2022a; Sohoni et al., 2020) have shown the capability to identify hidden spurious attributes via clustering on the feature space for debiased representation learning. Likewise, we take advantage of feature clustering for adaptive robust scaling; we obtain the optimal class-specific scaling factors based on the cluster membership of each sample. The overall algorithm of our instance-wise robust scaling (IRS) is described as follows.

1. Perform clustering with the validation dataset on the feature space and store the cluster centroids.
2. Find the optimal scaling factor for each cluster.
3. Apply the estimated scaling factor to each test example based on its cluster membership.

In step 1, we use a simple $K$-means clustering algorithm. Empirically, when $K$ is sufficiently large, $i.e.$, $K > 10$, IRS achieves stable and superior results, compared to the original class-specific scaling.

## 3.4 ROBUST COVERAGE

Although the robust scaling identifies the desired performance point on the trade-off curve, from the perspective of performance evaluation, it still reflects only a single point on the trade-off curve while

---

[1]Refer to our supplementary document for the coherency of robust scaling in the validation and test sets.

Table 1: Experimental results of the robust scaling (RS) and instance-wise robust scaling (IRS) on the CelebA dataset using ResNet-18 with the average of three runs (standard deviations in parentheses), where RS and IRS are applied to maximize each target metric independently. *Gain* indicates the average and standard deviation of performance improvement. '∗' indicates that the backbone method requires group supervision for training examples. When combined with existing debiasing approaches, RS maximizes all target metrics consistently and IRS further boosts the performance.

| Method | Robust Coverage | | Accuracy (%) | | | | | |
| --- | --- | --- | --- | --- | --- | --- | --- | --- |
| | Worst-group | Unbiased | Worst-group | (Gain) | Unbiased | (Gain) | Average | (Gain) |
| ERM | - | - | 34.5 (6.1) | - | 77.7 (1.8) | - | 95.5 (0.4) | - |
| ERM + RS | 83.0 (0.8) | 88.1 (0.6) | 82.8 (3.3) | +47.7 (7.8) | 91.2 (0.5) | +13.3 (2.0) | **95.8 (0.2)** | +0.4 (0.2) |
| ERM + IRS | **83.4 (0.1)** | **88.4 (0.4)** | **87.2 (2.0)** | +52.7 (3.3) | **91.7 (0.2)** | +13.8 (1.6) | 95.8 (0.1) | +0.4 (0.3) |
| CR | - | - | 70.6 (6.0) | - | 88.7 (1.2) | - | 94.2 (0.7) | - |
| CR + RS | 82.9 (0.5) | 88.2 (0.3) | 82.7 (5.2) | +12.2 (7.5) | 91.0 (1.0) | +2.2 (1.3) | 95.4 (0.5) | +1.3 (0.4) |
| CR + IRS | **83.6 (1.1)** | **88.6 (0.5)** | **84.8 (1.5)** | +14.2 (5.2) | **91.3 (0.4)** | +2.5 (1.4) | **95.5 (0.1)** | +1.3 (0.3) |
| SUBY | - | - | 65.7 (3.9) | - | 87.5 (0.9) | - | 94.5 (0.7) | - |
| SUBY + RS | 81.5 (1.0) | 87.4 (0.1) | 80.8 (2.9) | +15.1 (3.0) | 90.5 (0.8) | +3.0 (0.9) | 95.3 (0.6) | +0.8 (0.6) |
| SUBY + IRS | **82.3 (1.1)** | **87.8 (0.2)** | **82.3 (2.0)** | +16.5 (4.1) | **90.8 (0.8)** | +3.3 (1.1) | **95.5 (0.3)** | +1.1 (0.4) |
| LfF | - | - | 55.6 (6.6) | - | 81.5 (2.8) | - | 92.4 (0.8) | - |
| LfF + RS | 74.1 (3.5) | 79.7 (2.6) | 78.7 (4.1) | +23.2 (2.5) | 85.4 (2.4) | +4.0 (0.8) | **93.4 (0.7)** | +1.0 (0.2) |
| LfF + IRS | **74.6 (4.1)** | **79.8 (3.1)** | **78.9 (5.3)** | +23.4 (4.1) | **86.0 (2.2)** | +4.6 (1.5) | 93.1 (1.5) | +0.7 (0.5) |
| JTT | - | - | 75.1 (3.6) | - | 85.9 (1.4) | - | 89.8 (0.8) | - |
| JTT + RS | 77.3 (0.7) | 81.9 (0.7) | 82.9 (2.3) | +7.8 (3.0) | 87.6 (0.5) | +1.7 (0.4) | 90.3 (1.3) | +0.6 (0.1) |
| JTT + IRS | **78.9 (2.1)** | **82.1 (1.5)** | **84.9 (4.5)** | +9.8 (3.7) | **88.5 (0.8)** | +2.5 (0.8) | **91.0 (1.8)** | +1.2 (0.5) |
| GR∗ | - | - | 88.6 (1.9) | - | 92.0 (0.4) | - | 92.9 (0.8) | - |
| GR∗ + RS | 86.9 (0.4) | 88.4 (0.2) | 90.0 (1.6) | +1.4 (1.1) | 92.4 (0.5) | +0.5 (0.4) | 93.8 (0.4) | +0.8 (0.5) |
| GR∗ + IRS | **87.0 (0.2)** | **88.6 (0.2)** | **90.0 (2.3)** | +1.4 (1.8) | **92.6 (0.6)** | +0.6 (0.4) | **94.2 (0.3)** | +1.3 (1.0) |
| SUBG∗ | - | - - | 87.8 (1.2) | - | 90.4 (1.2) | - | 91.9 (0.3) | - |
| SUBG∗ + RS | 83.6 (1.6) | 87.5 (0.7) | 88.3 (0.7) | +0.5 (0.4) | 90.9 (0.5) | +0.5 (0.5) | 93.9 (0.2) | +1.9 (0.6) |
| SUBG∗ + IRS | **84.5 (0.8)** | **87.9 (0.1)** | **88.7 (0.6)** | +0.8 (0.7) | **91.0 (0.3)** | +0.6 (0.9) | **94.0 (0.2)** | +2.1 (1.0) |
| Group DRO∗ | - | - | 88.4 (2.3) | - | 92.0 (0.4) | - | 93.2 (0.8) | - |
| Group DRO∗ + RS | 87.3 (0.2) | 88.3 (0.2) | 89.7 (1.2) | +1.4 (1.0) | 92.3 (0.1) | +0.4 (0.2) | 93.9 (0.3) | +0.7 (0.5) |
| Group DRO∗ + IRS | **87.5 (0.4)** | **88.4 (0.2)** | **90.0 (2.3)** | +2.6 (1.8) | **92.6 (0.6)** | +0.6 (0.4) | **94.7 (0.3)** | +1.5 (1.1) |

ignoring all other possible Pareto optima. For a more comprehensive evaluation of an algorithm, we propose a convenient evaluation measure that yields a scalar summary of the robust-average accuracy trade-off. Formally, we define the *robust coverage* as

$$
\text{(Robust coverage)} := \int_{c=0}^{1} \max_{\mathbf{s}} \left\{ \text{RA}^{\mathbf{s}} | \text{AA}^{\mathbf{s}} \geq c \right\} dc \approx \sum_{d=0}^{D-1} \frac{1}{D} \max_{\mathbf{s}} \left\{ \text{RA}^{\mathbf{s}} | \text{AA}^{\mathbf{s}} \geq \frac{d}{D} \right\}, \quad (4)
$$

where $\text{RA}^{\mathbf{s}}$ and $\text{AA}^{\mathbf{s}}$ denote the robust and average accuracies and $D = 10^3$ is the number of slices for discretization. The robust coverage measures the area under the Pareto frontier of the robust-average accuracy trade-off curve, where the maximum operation in (4) finds the Pareto optimum for each threshold. We use either WA or UA as RA depending on the target objective of robust coverage in (3).

## 4 EXPERIMENTS

### 4.1 EXPERIMENTAL SETUP

**Implementation details**  Following prior works, we adopt ResNet-18, ResNet-50 (He et al., 2016), and DenseNet-121 (Huang et al., 2017), pretrained on ImageNet (Deng et al., 2009), as our backbone networks for the CelebA, Waterbirds, and FMoW-WILDS datasets, respectively. For the text classification dataset, CivilComments-WILDS, we use DistillBert (Sanh et al., 2019). We employ the standard $K$-means clustering for IRS, where the number of clusters is set to 20, *i.e.*, $K = 20$, for all experiments. We select the final model with the scaling factor that gives the best unbiased coverage in the validation split. Our implementations are based on the Pytorch (Paszke et al., 2019) framework and all experiments are conducted on a single unit of NVIDIA Titan XP GPU. We are planning to release our source codes. Please refer to our supplementary file for the details about the dataset usage.

**Evaluation metrics**  We evaluate all algorithms in terms of the proposed unbiased and worst-group coverages for comprehensive evaluation, and additionally use the average, unbiased, and worst-group

Table 2: Experimental results of RS and IRS on the Waterbirds dataset using ResNet-50 with the average and standard deviation of three runs, where RS and IRS are applied to maximize each target metric independently. RS and IRS improve both robust and average accuracies consistently.

| | Robust Coverage | | Accuracy (%) | | | | | |
|---|---|---|---|---|---|---|---|---|
| Method | Worst-group | Unbiased | Worst-group | (Gain) | Unbiased | (Gain) | Average | (Gain) |
| ERM | - | - | 76.3 (0.8) | - | 89.4 (0.6) | - | 97.2 (0.2) | - |
| ERM + RS | 76.1 (1.4) | 82.6 (1.3) | 81.6 (1.9) | +5.3 (1.3) | 89.8 (0.5) | +0.4 (0.4) | 97.5 (0.1) | +0.4 (0.2) |
| ERM + IRS | **83.4 (1.1)** | **86.9 (0.4)** | **89.3 (0.5)** | +13.0 (0.9) | **92.7 (0.4)** | +3.3 (0.7) | **97.5 (0.3)** | +0.3 (0.4) |
| CR | - | - | 76.1 (0.7) | - | 89.1 (0.7) | - | 97.1 (0.3) | - |
| CR + RS | 73.6 (2.3) | 82.0 (1.5) | 79.4 (2.4) | +3.4 (1.8) | 89.4 (1.0) | +0.3 (0.4) | **97.5 (0.3)** | +0.4 (0.1) |
| CR + IRS | **84.2 (2.5)** | **88.3 (1.0)** | **88.2 (2.7)** | +12.2 (2.1) | **92.1 (0.7)** | +3.1 (0.1) | 97.4 (0.2) | +0.3 (0.2) |
| SUBY | - | - | 72.8 (4.1) | - | 84.9 (0.4) | - | 93.8 (1.5) | - |
| SUBY + RS | 72.5 (1.0) | 81.2 (1.4) | 75.9 (4.4) | +3.4 (1.8) | 86.3 (0.9) | +2.3 (0.9) | 95.5 (0.2) | +1.7 (1.1) |
| SUBY + IRS | **78.8 (2.7)** | **85.9 (1.0)** | **82.1 (4.0)** | +9.3 (1.1) | **89.1 (0.9)** | +4.2 (1.0) | **96.2 (0.6)** | +2.4 (1.4) |
| GR* | - | - | 86.1 (1.3) | - | 89.3 (0.9) | - | 95.1 (1.3) | - |
| GR* + RS | 83.7 (0.3) | 86.8 (0.7) | **89.3 (1.3)** | +3.2 (2.0) | 92.0 (0.7) | +2.7 (1.3) | 95.4 (1.3) | +0.4 (0.2) |
| GR* + IRS | **84.8 (1.7)** | **87.4 (0.4)** | 89.1 (0.8) | +3.0 (1.6) | **92.2 (1.0)** | +2.9 (1.6) | **95.6 (0.8)** | +0.6 (0.3) |
| SUBG* | - | - | 86.5 (0.9) | - | 88.2 (1.2) | - | 87.3 (1.1) | - |
| SUBG* + RS | 80.6 (2.0) | 82.3 (2.0) | 87.1 (0.7) | +0.6 (0.5) | **88.5 (1.2)** | +0.3 (0.3) | 91.3 (0.4) | +4.0 (0.9) |
| SUBG* + IRS | **82.2 (0.8)** | **84.1 (0.8)** | **87.3 (1.3)** | +0.8 (0.6) | 88.2 (1.2) | +0.0 (0.2) | **93.5 (0.4)** | +6.2 (1.5) |
| Group DRO* | - | - | 88.0 (1.0) | - | 92.5 (0.9) | - | 95.8 (1.8) | - |
| Group DRO* + RS | 83.4 (1.1) | 87.4 (1.4) | 89.1 (1.7) | +1.1 (0.8) | 92.7 (0.8) | +0.2 (0.1) | 96.4 (1.5) | +0.5 (0.5) |
| Group DRO* + IRS | **86.3 (2.3)** | **90.1 (2.6)** | **90.8 (1.3)** | +2.8 (1.5) | **93.9 (0.2)** | +1.4 (0.9) | **97.1 (0.4)** | +1.2 (0.8) |

Table 3: Experimental results on the CivilComments-WILDS dataset using DistilBert. RS and IRS achieve consistent performance gains in all settings.

| | Robust Coverage | | Accuracy (%) | | | | | |
|---|---|---|---|---|---|---|---|---|
| Method | Worst-group | Unbiased | Worst-group | (Gain) | Unbiased | (Gain) | Average | (Gain) |
| ERM | - | - | 54.5 (6.8) | - | 75.0 (1.2) | - | 92.3 (0.4) | - |
| ERM + RS | 57.2 (5.1) | 70.9 (1.5) | 65.5 (1.2) | +11.0 (2.5) | 78.6 (1.5) | +3.7 (2.4) | **92.5 (0.3)** | +0.2 (0.1) |
| ERM + IRS | **59.2 (5.3)** | **71.2 (2.3)** | **67.0 (2.3)** | +12.5 (2.7) | **78.8 (1.1)** | +3.8 (1.7) | **92.5 (0.3)** | +0.2 (0.1) |
| GR* | - | - | 64.7 (1.1) | - | 78.4 (0.2) | - | 87.2 (1.0) | - |
| GR* + RS | 59.0 (2.8) | 69.8 (1.0) | 66.0 (0.5) | +1.3 (0.6) | 78.5 (0.1) | +0.1 (0.1) | 87.9 (0.8) | +0.7 (0.3) |
| GR* + IRS | **59.7 (1.6)** | **70.1 (0.7)** | **66.2 (0.4)** | +1.6 (0.7) | **78.6 (0.1)** | +0.2 (0.2) | **88.4 (0.6)** | +1.2 (0.6) |
| Group DRO* | - | - | 67.7 (0.6) | - | 78.4 (0.6) | - | 90.0 (0.1) | - |
| Group DRO* + RS | 60.6 (0.6) | 71.5 (0.3) | 68.8 (0.7) | +1.1 (0.5) | **78.8 (0.4)** | +0.4 (0.3) | 90.5 (0.2) | +0.5 (0.3) |
| Group DRO* + IRS | **62.1 (0.7)** | **71.9 (0.2)** | **69.6 (0.4)** | +1.9 (0.6) | **78.8 (0.5)** | +0.4 (0.6) | **90.8 (0.3)** | +0.8 (0.3) |

accuracies for comparisons. Following previous works (Sagawa et al., 2020; Liu et al., 2021), we report the adjusted average accuracy instead of the naïve version for the Waterbirds dataset due to its dataset imbalance issue; we first calculate the accuracy for each group and then report the weighted average, where the weights are given by the relative portion of each group in the training set. We ran the experiments three times for each algorithm and report their average and standard deviation.

## 4.2 RESULTS

**CelebA** Table 1 presents the experimental results of our robust scaling methods (RS and IRS) on top of the existing approaches including CR, SUBY, LfF, JTT, Group DRO*, GR*, and SUBG*[2] on the CelebA dataset, where '*' indicates the method that requires the group supervision in training sets. In this evaluation, RS and IRS choose scaling factors to maximize individual target metrics— worst-group, unbiased, and average accuracies[3]. As shown in the table, our robust scaling strategies consistently improve the performance for all target metrics. In terms of the robust coverage and robust accuracy after scaling, LfF and JTT are not superior to ERM on the CelebA dataset although their robust accuracies without scaling are much higher than ERM. The methods that leverage group supervision such as Group DRO and GR achieve better robust coverage results than the others, which verifies that group supervision helps to improve overall performance. For the group-supervised methods, our scaling technique achieves relatively small performance gains in robust accuracy since the gaps between robust and average accuracies are small and the original results are already close to

---

[2]A brief introduction to these methods is provided in the supplementary document.

[3]Since our robust scaling strategy is a simple post-processing method, we do not need to retrain models for each target measure and the cost is negligible, taking only a few seconds for each target metric.

Table 4: Experimental results on the FMoW-WILDS dataset using DenseNet-121. RS and IRS are effective even when training, validation, and test splits involve substantial domain shifts and the number of classes is large.

| Method | Robust Coverage | | Accuracy (%) | | | | | |
|---|---|---|---|---|---|---|---|---|
| | Worst-group | Unbiased | Worst-group | (Gain) | Unbiased | (Gain) | Average | (Gain) |
| ERM | - | - | 34.5 (1.4) | - | 51.7 (0.5) | - | 52.6 (0.8) | - |
| ERM + RS | 32.9 (0.4) | 39.4 (1.3) | 35.7 (1.6) | +1.2 (0.4) | 52.3 (0.3) | +0.6 (0.3) | 53.1 (0.8) | +0.6 (0.3) |
| ERM + IRS | **35.1 (0.2)** | **40.2 (1.1)** | **36.2 (1.4)** | +1.7 (0.3) | **52.4 (0.2)** | +0.7 (0.4) | **53.4 (0.9)** | +0.8 (0.4) |
| GR* | - | - | 31.4 (1.1) | - | 49.0 (0.9) | - | 50.1 (1.3) | - |
| GR* + RS | 30.2 (1.2) | 37.7 (0.6) | 35.5 (0.4) | +4.2 (0.7) | 49.8 (0.7) | +0.8 (0.3) | 50.7 (1.2) | +0.6 (0.1) |
| GR* + IRS | **31.7 (1.0)** | **38.9 (2.1)** | **35.7 (0.9)** | +4.4 (0.4) | **50.1 (0.6)** | +1.1 (0.3) | **50.8 (1.4)** | +0.7 (0.1) |
| Group DRO* | - | - | 33.7 (2.0) | - | 50.4 (0.7) | - | 52.0 (0.4) | - |
| Group DRO* + RS | 30.8 (1.8) | 38.2 (0.7) | 36.0 (2.4) | +2.3 (0.4) | 50.9 (0.6) | +0.4 (0.4) | 52.4 (0.2) | +0.5 (0.2) |
| Group DRO* + IRS | **34.1 (0.8)** | **40.7 (0.5)** | **36.4 (2.3)** | +2.7 (0.4) | **51.1 (0.3)** | +0.7 (0.5) | **52.7 (0.2)** | +0.7 (0.2) |

the optimal robust accuracy. Note that, compared to RS, IRS further boosts the robust coverage and all types of accuracies consistently in all algorithms.

**Waterbirds** Table 2 demonstrates the outstanding performance of our approaches with all baselines on the Waterbirds dataset. Among the compared algorithms, GR and SUBG are reweighting and subsampling methods based on group frequency, respectively. Although the two baseline approaches exhibit competitive robust accuracy, the average accuracy of SUBG is far below than GR (87.3% vs. 95.1%). This is mainly because SUBG drops a large portion of training samples (95%) to make all groups have the same size, resulting in the significant loss of average accuracy. Subsampling generally helps to achieve high robust accuracy, but it degrades the overall trade-off as well as the average accuracy, consequently hindering the benefits of robust scaling. This observation is coherent to our main claim; the optimization towards the robust accuracy is incomplete and more comprehensive evaluation criteria are required to understand the exact behavior of debiasing algorithms. Note that GR outperforms SUBG in terms of all accuracies after adopting the proposed RS or IRS.

**CivilComments-WILDS** We also validate the effectiveness of the proposed approach in a large-scale text classification dataset, CivilComments-WILDS (Koh et al., 2021), which has 8 attribute groups. As shown in Table 3, our robust scaling strategies still achieve meaningful performance improvements for all baselines on this dataset. Although group-supervised baselines such as GR and Group DRO accomplish higher robust accuracies than the ERM without scaling, ERM benefits from RS and IRS greatly. ERM+IRS outperforms both Group DRO and GR in average accuracy while achieving competitive worst-group and unbiased accuracies, even without group supervision in training samples and extra training.

**FMoW-WILDS** FMoW-WILDS (Koh et al., 2021) is a high-resolution satellite imagery dataset with 65 classes and 5 attribute groups, which involves domain shift issues as train, validation, and test splits come from different years. We report the results from our experiments in Table 4, which shows that GR and Group DRO have inferior performance even compared with ERM. On the other hand, our robust scaling methods do not suffer from any performance degradation and even enhance all kinds of accuracies substantially. This fact supports the strengths and robustness of our framework in more challenging datasets with distribution shifts.

## 4.3 ANALYSIS

**Validation set sizes** We analyze the impact of the validation set size on the robustness of our algorithm. Table 5 presents the ERM results on the CelebA dataset by varying the validation set size to $\{100\%, 50\%, 10\%, 1\%\}$ of its full size. Note that other approaches also require validation sets with group annotations for early stopping and hyperparameter tuning, which are essential to achieve high robust accuracy. As shown in the table, with only $10\%$ or $50\%$ of the validation set, both RS and IRS achieve almost equivalent performance to the versions with the entire validation set. Surprisingly, even only $1\%$ of the validation set is enough for RS to gain sufficiently high robust accuracy but inevitably entails a large variance of results. On the other hand, IRS suffers from performance degradation when only $1\%$ of the validation set is available. This is mainly because IRS takes advantage of feature clustering on the validation set, which would need more examples for stable results. In overall, our robust scaling strategies generally improve performance substantially even with a limited number of validation examples with group annotations for all cases.

Table 5: Ablation study on the size of validation set in our robust scaling strategies on CelebA.

| Method | Validation set size | Worst-group Acc. | Gain | Unbiased Acc. | Gain |
|--------|--------------------|-----------------|------|---------------|------|
| ERM | - | 34.5 (6.1) | - | 77.7 (1.8) | - |
| + RS | 100% | 82.8 (3.3) | **+48.3** | 91.2 (0.5) | **+13.5** |
| + RS | 50% | 83.3 (3.7) | **+48.8** | 91.5 (0.9) | **+13.8** |
| + RS | 10% | 82.4 (4.3) | **+48.0** | 91.4 (0.8) | **+13.7** |
| + RS | 1% | 79.2 (10.3) | **+44.7** | 90.8 (2.2) | **+13.1** |
| + IRS | 100% | 88.7 (0.9) | **+54.2** | 92.0 (0.3) | **+14.3** |
| + IRS | 50% | 86.9 (2.0) | **+52.4** | 91.8 (0.4) | **+14.1** |
| + IRS | 10% | 84.4 (6.3) | **+50.0** | 91.4 (1.0) | **+13.7** |
| + IRS | 1% | 60.4 (14.4) | **+25.9** | 85.8 (3.2) | **+8.0** |

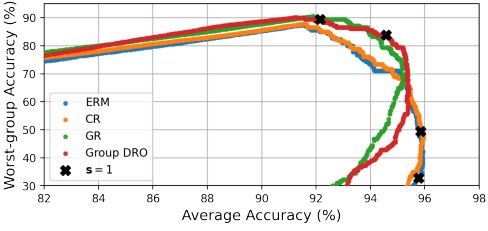 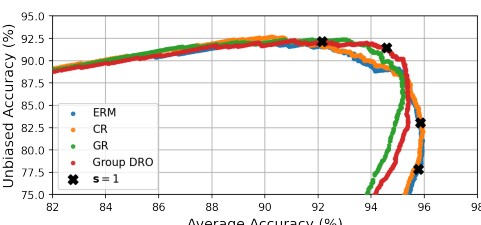

(a) With the worst-group accuracy          (b) With the unbiased accuracy

Figure 4: The robust-average accuracy trade-off curves on the CelebA dataset.

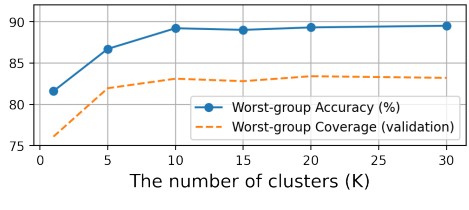 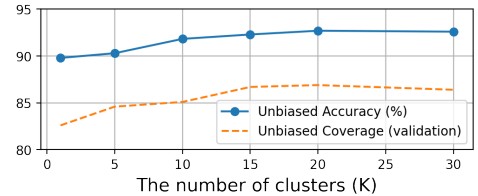

(a) With the worst-group accuracy          (b) With the unbiased accuracy

Figure 5: Sensitivity analysis with respect to the number of clusters in IRS on Waterbirds. The tendency of the robust coverage in the validation split (orange) is similar with the robust accuracy in the test split (blue).

**Accuracy trade-off**    Figure 4 depicts the robust-average accuracy trade-offs of several existing algorithms on the CelebA dataset. The black markers denote the points without scaling, implying that there is room for improvement in robust accuracy along the trade-off curve.

**Number of clusters**    We adjust the number of clusters for feature clustering in IRS on the Waterbirds dataset. Figure 5 illustrates that the worst-group and unbiased accuracies gradually improve as $K$ increases and are stable with a sufficiently large $K(> 10)$. The leftmost point ($K = 1$) denotes RS in each figure. We also plot the robust coverage results in the validation split, which are almost consistent with the robust accuracy measured in the test dataset.

## 5  CONCLUSION

We presented a simple but effective post-processing method that provides a novel perspective of group robustness. Our work starts from the observation that there exists a clear trade-off between robust and average accuracies in existing works. From this observation, we first proposed the robust scaling strategy, which captures the full trade-off landscape and identifies any desired performance point on the trade-off curve with no extra training. Moreover, we proposed an instance-wise robust scaling algorithm with adaptive scaling factors, which is effective to enhance the trade-off itself. Based on these strategies, we introduced a novel convenient measure that summarizes the trade-off from a Pareto optimal perspective for a comprehensive evaluation of group robustness. We believe that our approaches are helpful for analyzing the exact behavior of existing debiasing methods and paving the way in the future research direction.

**Ethics Statement** Addressing the potential risks caused by dataset or algorithmic bias is a timely subject, particularly given the pervasiveness of AI in our daily lives. Although existing approaches have apparently improved robust accuracy and seem to successfully mitigate these risks, it's important to note that such gains mainly come from the trade-offs at the expense of average accuracy. By considering the trade-offs comprehensively, we believe our framework can actually help to measure these potentially harmful risks and pave the way in the future research direction. We also acknowledge that we have read and commit to adhering to the ICLR code of ethics.

**Reproducibility Statement** We provide the details about the dataset usages, implementation, evaluation metrics, and training environments in Section 4.1 and C. We report all experimental results in the main tables based on the average of three runs with standard deviation.

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

# A    COMPARISONS

Below is a brief introduction of the comparisons used in our experiments.

**ERM**    Given a loss function $\ell(\cdot)$, the objective of empirical risk minimization is optimizing the following loss over training data:

$$\min_{\theta} \left\{ \frac{1}{n} \sum_{i=1}^{n} \ell(f_{\theta}(x_i), y_i) \right\}. \tag{5}$$

**Class reweighting (CR)**    To mitigate the class imbalance issue, we can simply reweight the samples based on the inverse of class frequency in the training split,

$$\min_{\theta} \left\{ \frac{1}{n} \sum_{i=1}^{n} \omega_i \ell(f_{\theta}(x_i), y_i) \right\} \text{ where } \omega_i = \frac{n}{\sum_j \mathbb{1}(y_j = y_i)}. \tag{6}$$

**LfF**    Motivated by the observation that bias-aligned samples are more easily learned, LfF (Nam et al., 2020) simultaneously trains a pair of neural network $(f_B, f_D)$. The biased model $f_B$ is trained with generalized cross-entropy loss which intends to amplify bias, while the debiased model $f_D$ is trained with a standard cross-entropy loss, where each sample $(x_i, y_i)$ is reweighted by the following relative difficulty score:

$$\omega_i = \frac{\ell(f_{\theta}^{B}(x_i), y_i)}{\ell(f_{\theta}^{B}(x_i), y_i) + \ell(f_{\theta}^{D}(x_i), y_i)}. \tag{7}$$

**JTT**    JTT (Liu et al., 2021) consists of two-stage procedures. In the first stage, JTT trains a standard ERM model $\hat{f}(\cdot)$ for several epochs and identifies an error set $E$ of training examples that are misclassified:

$$E := \{(x_i, y_i) \text{ s.t. } \hat{f}(x_i) \neq y_i\}. \tag{8}$$

Next, they train a final model $f_{\theta}(\cdot)$ by upweighting the examples in the error set $E$ as

$$\min_{\theta} \left\{ \lambda_{\text{up}} \sum_{(x,y) \in E} \ell(f_{\theta}(x), y) + \sum_{(x,y) \notin E} \ell(f_{\theta}(x), y) \right\}. \tag{9}$$

**Group DRO**    Group DRO (Sagawa et al., 2020) aims to minimize the empirical worst-group loss formulated as:

$$\min_{\theta} \left\{ \max_{g \in \mathcal{G}} \frac{1}{n_g} \sum_{i|g_i=g}^{n_g} \ell(f_{\theta}(x_i), y_i) \right\} \tag{10}$$

where $n_g$ is the number of samples assigned to $g^{\text{th}}$ group. Unlike previous approaches, group DRO requires group annotations $g = (y, a)$ on the training split.

**Group reweighting (GR)**    Using group annotations, we can extend class reweighting method to group reweighting one based on the inverse of group frequency in the training split, *i.e.*,

$$\min_{\theta} \left\{ \frac{1}{n} \sum_{i=1}^{n} \omega_i \ell(f_{\theta}(x_i), y_i) \right\}$$

$$\text{where } \omega_i = \frac{n}{\sum_j \mathbb{1}(y_j = y_i, a_j = a_i)} \tag{11}$$

**SUBY/SUBG**    To mitigate the data imbalance issue, SUBY subsample majority classes, so all classes have the same size with the smallest class on the training dataset, as in Idrissi et al. (2022). Similarly, SUBG subsample majority groups.

Table A6: Results of the attribute-specific robust scaling (ARS) on the CelebA and Waterbirds datasets with the average of three runs (standard deviations in parenthesis), where ARS is applied to maximize each target metric independently. Note that our post-processing strategy, ARS, allows ERM to achieve competitive performance to Group DRO that utilizes the group supervision during training.

| Dataset | Method | Robust Coverage | | Accuracy (%) | | |
| | | Worst. | Unbiased | Worst. | Unbiased | Average |
| --- | --- | --- | --- | --- | --- | --- |
| | ERM | - | - | 34.5 (6.1) | 77.7 (1.8) | 95.5 (0.4) |
| CelebA | ERM + ARS | **87.6 (1.0)** | **89.0 (0.2)** | **88.5 (1.8)** | 91.9 (0.3) | **95.8 (0.1)** |
| | Group DRO | 87.3 (0.2) | 88.3 (0.2) | 88.4 (2.3) | **92.0 (0.4)** | 93.2 (0.8) |
| | ERM | - | - | 76.3 (0.8) | 89.4 (0.6) | 97.2 (0.2) |
| Waterbirds | ERM + ARS | **84.4 (1.9)** | **87.8 (1.7)** | **89.3 (0.4)** | 92.5 (0.4) | **97.5 (1.0)** |
| | Group DRO | 83.4 (1.1) | 87.4 (2.3) | 88.0 (1.0) | **92.5 (0.9)** | 95.8 (1.8) |

## B  ATTRIBUTE-SPECIFIC ROBUST SCALING WITH GROUP SUPERVISION

If the supervision of group (spurious-attribute) information can be utilized during our robust scaling, it will provide flexibility to further improve the performance. To this end, we first partition the examples based on the values of spurious attributes and find the optimal scaling factors for each partition separately. Like as the original robust scaling procedure, we obtain the optimal scaling factors for each partition in the validation split and apply them to the test split. However, this partition-wise scaling is basically unavailable because we do not know the spurious attribute values of the examples in the test split and thus cannot partition them, In other words, we need to estimate the spurious-attribute values in the test split for partitioning. To conduct attribute-specific robust scaling (ARS), we follow a simple algorithm described below:

1. Partition the examples in the validation split by the values of the spurious attribute.
2. Find the optimal scaling factors for each partition in the validation split.
3. Train an independent estimator model to classify spurious attribute.
4. Estimate the spurious attribute values of the examples in the test split using the estimator, and partition the test samples according to their estimated spurious attribute values.
5. For each sample in the test split, apply the optimal scaling factors obtained in step 2 based on its partition.

To find a set of scale factors corresponding to each partition, we adopt a naïve greedy algorithm that performed in one partition at a time. This attribute-specific robust scaling further increases the robust accuracy compared to the original robust scaling, and also improves the robust coverage, as shown in Table A6. Note that our attribute-specific scaling strategy allows ERM to match the supervised state-of-the-art approach, Group DRO (Sagawa et al., 2020).

One limitation is that it requires the supervision of spurious attribute information to train the estimator model in step 3. However, we notice that only a very few examples with the supervision is enough to train the spurious-attribute estimator, because it is much easier to learn as the word "spurious correlation" suggests. To determine how much the group-labeled data is needed, we train several spurious-attribute estimators by varying the number of group-labeled examples, and conduct ARS using the estimators. Table A7 validates that, compared to the overall training dataset size, a very small amount of group-labeled examples is enough to achieve high robust accuracy.

## C  EXPERIMENTAL DETAILS

### C.1  DATASETS

CelebA (Liu et al., 2015) is a large-scale dataset for face image recognition, consisting of 202,599 celebrity images, with 40 attributes labeled on each image. Among the attributes, we primarily examine *hair color* and *gender* attributes as a target and spurious attributes, respectively. We follow the original train-validation-test split (Liu et al., 2015) for all experiments in the paper.

Table A7: Effects of the size of group-labeled examples on the attribute-specific robust scaling on the CelebA dataset. Group-labeled size denotes a ratio of group-labeled samples among all training examples for training estimators. Spurious accuracy indicates the average accuracy of spurious-attribute classification using the estimators on the test split.

| Group-labeled size | Accuracy (%) Spurious | Accuracy (%) Worst-group | Unbiased | Average | Robust Coverage Worst-group | Unbiased |
|---|---|---|---|---|---|---|
| 100% | 98.4 | 89.1 (3.0) | 92.4 (1.1) | 93.1 (1.2) | 87.6 (1.0) | 89.0 (0.5) |
| 10% | 97.7 | 88.5 (1.8) | 91.9 (0.3) | 92.8 (0.6) | 86.8 (0.4) | 89.0 (0.2) |
| 1% | 95.8 | 88.5 (1.8) | 91.9 (0.3) | 92.9 (0.6) | 87.1 (0.3) | 89.0 (0.2) |
| 0.1% | 92.6 | 88.4 (2.1) | 91.8 (0.5) | 92.4 (0.8) | 87.1 (0.3) | 89.0 (0.2) |

Table A8: Realized robust coverage results on the Waterbirds and CelebA datasets with the average of three runs (standard deviations in parenthesis).

| Dataset | Method | Robust Coverage Worst-group | Unbiased | Realized Robust Coverage Worst-group | Unbiased |
|---|---|---|---|---|---|
| Waterbirds | ERM | 70.3 (1.3) | 79.4 (0.7) | 69.0 (1.5) | 78.7 (0.8) |
| Waterbirds | CR | 68.9 (1.1) | 78.5 (0.5) | 67.8 (1.2) | 77.9 (0.4) |
| Waterbirds | Group DRO | 80.8 (0.6) | 85.2 (0.1) | 78.6 (1.0) | 83.8 (0.4) |
| Waterbirds | GR | 78.8 (5.6) | 83.7 (0.7) | 77.9 (1.4) | 82.8 (0.8) |
| CelebA | ERM | 78.9 (1.7) | 86.0 (0.6) | 75.9 (2.2) | 85.4 (0.7) |
| CelebA | CR | 77.2 (2.8) | 85.6 (0.9) | 71.8 (1.3) | 85.0 (0.6) |
| CelebA | Group DRO | 84.2 (0.6) | 86.7 (0.5) | 81.0 (1.7) | 86.1 (0.2) |
| CelebA | GR | 84.2 (0.5) | 87.5 (0.3) | 81.2 (1.6) | 87.0 (0.5) |

Waterbirds (Sagawa et al., 2020) is a synthesized dataset, which are created by combining bird images in the CUB dataset (Wah et al., 2011) and background images from the Places dataset (Zhou et al., 2017), consisting of 4,795 training examples. The two attributes—one is the type of bird, {waterbird, landbird} and the other is background places, {water, land}, are used for the experiments with this dataset. CivilComments-WILDS (Koh et al., 2021) is a large-scale text dataset, which has 269,038 training comments, 45,180 validation comments, and 133,782 test comments. This task is to classify whether an online comment is toxic or not, which is spuriously correlated to demographic identities (*male, female, White, Black, LGBTQ, Muslim, Christian, and other religion*). FMoW-WILDS (Koh et al., 2021) is based on the Functional Map of the World dataset (Christie et al., 2018), comprising high-resolution satellite images from over 200 countries and over the years 2002-2018. The label is one of 62 building or land use categories, and the attribute represents both the year and geographical regions (*Africa, the Americas, Oceania, Asia, or Europe*). It consists of 76,863 training images from the years 2002-2013, 19,915 validation images from the years 2013-2016, and 22,108 test images from the years 2016-2018.

## C.2 CLASS-SPECIFIC SCALING

To identify the optimal points, we obtain a set of the average and robust accuracy pairs using a wide range of the class-specific scaling factors, *i.e.*, $\mathbf{s}_i = (1.05)^n$ for $-200 \le n \le 200$ for $i^{\text{th}}$ class. Note that we search for the scaling factor of each class in a greedy manner, as stated in Section 3.2.

## C.3 HYPERPARAMETER TUNING

We tune the learning rate in $\{10^{-3}, 10^{-4}, 10^{-5}, 10^{-6}\}$ and the weight decay in $\{1.0, 0.1, 10^{-2}, 10^{-4}\}$ for all baselines on all datasets. We used 0.5 of $q$ for LfF. For JTT, we searched $\lambda_{\text{up}}$ in $\{20, 50, 100\}$ and updated the error set every epoch for CelebA dataset and every 60 epochs for Waterbirds dataset. For Group DRO, we tuned $C$ in $\{0, 1, 2, 3, 4\}$, and used 0.1 of $\eta$.

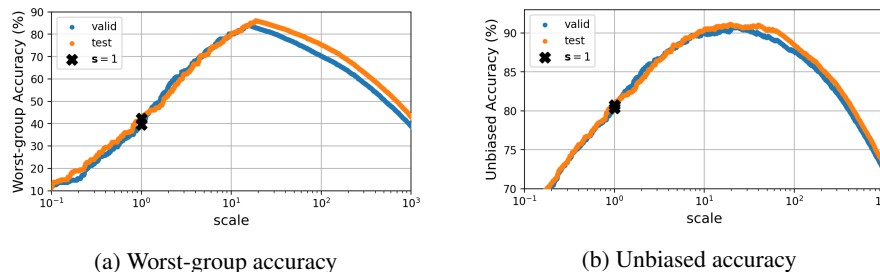

(a) Worst-group accuracy

(b) Unbiased accuracy

Figure A6: Effects of varying the class-specific scaling factors on the robust accuracy using ERM model on the CelebA dataset. Since this experiment is based on the binary classifier, a single scaling factor is varied with the other fixed to one. These results show that the optimal scaling factor identified in the validation set can be used in the test set to get the final robust prediction.

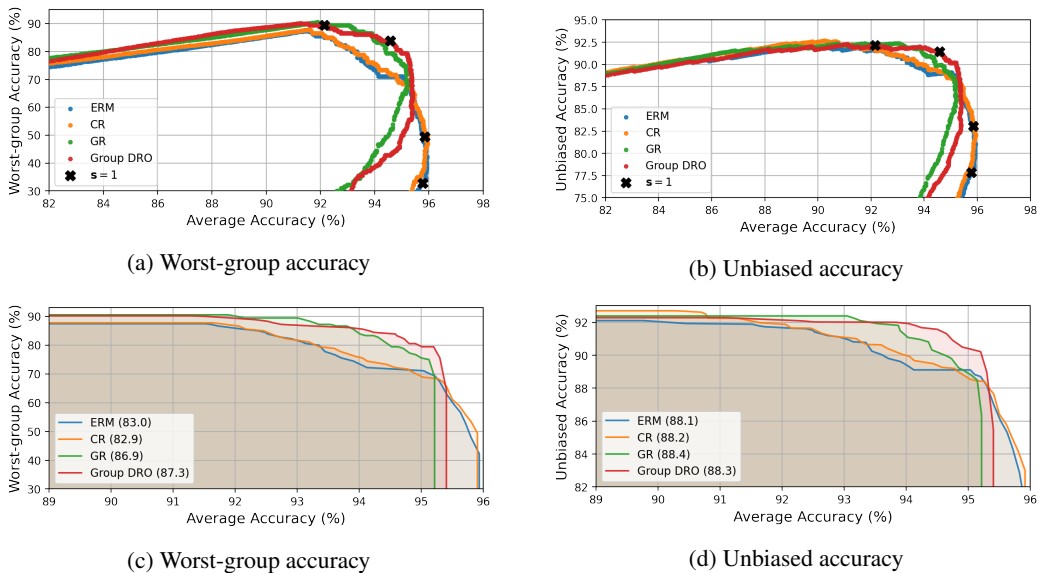

(a) Worst-group accuracy

(b) Unbiased accuracy

(c) Worst-group accuracy

(d) Unbiased accuracy

Figure A7: The robust-average accuracy trade-off curves ((a), (b)) and their corresponding robust coverage curves ((c), (d)), respectively, on the CelebA dataset. The curves in (c) and (d) represent the Pareto frontiers of the curves in (a) and (b), respectively. In (c) and (d), the numbers in the legend denote the robust coverage, which measures the area under the curve.

## D    ADDITIONAL RESULTS

**Feasibility**    We visualize the relationship between scaling factors and robust accuracies in Figure A6, where the curves are constructed based on validation and test splits are sufficiently well-aligned to each other. This implies that the optimal scaling factor identified in the validation set can be used in the test set to get the final robust prediction.

**Robust coverage curve**    Figure A7a and A7b are robust-average accuracy trade-off curves while Figure A7c and A7d are their corresponding robust coverage curves, which represent the Pareto frontiers of Figure A7a and A7b, respectively. The area under the curve in Figure A7c and A7d indicates the robust coverage of each algorithm.

**Scalability**    As mentioned in Section 3.2, we search for the scaling factor of each class in a greedy manner. Hence, the time complexity increases linearly with respect to the number of classes instead of the exponential growth with the full grid search; even with 1,000 classes, the whole process takes less than a few minutes in practice, which is negligible compared to the model training time. Moreover, we can reduce the computational cost even further by introducing the superclass concept

Table A9: Variations of robust scaling methods and their performances tested on the FairFace dataset.

| Method | Cost | Worst-group | Unbiased | Average |
|---|---|---|---|---|
| ERM | – | 15.8 | 47.0 | 54.1 |
| + RS (2 super classes) | $\mathcal{O}(n)$ | 18.6 | 51.8 | 52.9 |
| + RS (greedy search) | $\mathcal{O}(n)$ | **19.2** | 52.3 | **53.3** |
| + RS (full grid search) | $\mathcal{O}(n^9)$ | 19.0 | **52.8** | 53.1 |

Table A10: Results of our robust scaling methods on top of various baselines on the CelebA dataset, which supplement Table 1. Blue color denotes the target metric that the robust scaling aims to maximize. Compared to RS, IRS improves the overall trade-off.

| Method | Robust Coverage | | Accuracy (%) | | | Accuracy (%) | | |
|---|---|---|---|---|---|---|---|---|
| | Worst-group | Unbiased | Worst-group | Unbiased | Average | Worst-group | Unbiased | Average |
| ERM | - | - | 34.5 (6.1) | 77.7 (1.8) | **95.5 (0.4)** | 34.5 (6.1) | 77.7 (1.8) | 95.5 (0.4) |
| ERM + RS | 83.0 (0.7) | 88.1 (0.5) | 82.1 (3.7) | 91.1 (0.6) | 92.2 (1.3) | **45.0 (7.4)** | **81.7 (1.8)** | **95.8 (0.2)** |
| ERM + IRS | **83.4 (0.1)** | **88.4 (0.4)** | **87.2 (2.0)** | **91.7 (0.2)** | 91.5 (0.8) | 44.1 (4.2) | 81.3 (0.8) | **95.8 (0.1)** |
| CR | - | - | 70.6 (6.0) | 88.7 (1.2) | **94.2 (0.7)** | 70.6 (6.0) | 88.7 (1.2) | 94.2 (0.7) |
| CR + RS | 82.9 (0.5) | 88.2 (0.3) | 82.7 (5.2) | 91.0 (1.0) | 91.7 (1.3) | 48.5 (8.9) | 82.5 (2.2) | **95.8 (0.1)** |
| CR + IRS | **83.6 (1.1)** | **88.6 (0.5)** | **84.8 (1.5)** | **91.3 (0.4)** | 90.7 (1.3) | 48.8 (9.1) | 82.7 (2.4) | **95.8 (0.1)** |
| SUBY | - | - | 65.7 (3.9) | 87.5 (0.9) | **94.5 (0.7)** | 65.7 (3.9) | **87.5 (0.9)** | 94.5 (0.7) |
| SUBY + RS | 81.5 (1.0) | 87.4 (0.1) | 80.8 (2.9) | 90.5 (0.8) | 91.1 (1.7) | 45.4 (6.7) | 81.4 (2.0) | **95.5 (0.0)** |
| SUBY + IRS | **82.3 (1.1)** | **87.8 (0.2)** | **82.3 (2.0)** | **90.8 (0.8)** | 90.7 (1.9) | 46.0 (6.9) | 81.5 (2.1) | **95.5 (0.1)** |
| SUBG | - | - | 87.8 (1.2) | 90.4 (1.2) | **91.9 (0.3)** | 87.8 (1.2) | 90.4 (1.2) | 91.9 (0.3) |
| SUBG + RS | 83.6 (1.6) | 87.5 (0.7) | 88.3 (0.7) | 90.9 (0.5) | 90.6 (1.0) | 67.8 (6.5) | 85.2 (2.0) | 93.9 (0.2) |
| SUBG + IRS | **84.5 (0.8)** | **87.9 (0.1)** | **88.7 (0.6)** | **91.0 (0.3)** | 90.6 (0.8) | 68.5 (6.5) | 85.5 (1.9) | **94.0 (0.2)** |
| GR | - | - | 88.6 (1.9) | 92.0 (0.4) | **92.9 (0.8)** | 88.6 (1.9) | **92.0 (0.4)** | 92.9 (0.8) |
| GR + RS | 86.9 (0.4) | 88.4 (0.2) | **90.0 (1.6)** | 92.4 (0.5) | 92.5 (0.5) | 66.5 (0.3) | 85.4 (0.4) | 93.8 (0.4) |
| GR + IRS | **87.0 (0.2)** | **88.6 (0.2)** | 90.0 (2.3) | **92.6 (0.6)** | 92.5 (0.4) | 62.0 (5.3) | 84.5 (0.7) | **94.2 (0.3)** |
| GroupDRO | - | - | 88.4 (2.3) | 92.0 (0.4) | **93.2 (0.8)** | 88.4 (2.3) | **92.0 (0.4)** | 93.2 (0.8) |
| GroupDRO + RS | 87.3 (0.2) | 88.3 (0.2) | 89.7 (1.2) | 92.3 (0.1) | **93.7 (0.5)** | 64.9 (3.3) | 85.1 (0.7) | 93.9 (0.3) |
| GroupDRO + IRS | **87.5 (0.4)** | **88.4 (0.2)** | **90.0 (2.3)** | **92.6 (0.6)** | 93.5 (0.4) | 60.4 (5.4) | 84.4 (0.6) | **94.7 (0.3)** |

and allocating a single scaling factor for each superclass. We compare three different options—greedy search, superclass-level search, and full grid search—on the FairFace dataset (Kärkkäinen & Joo, 2021) with 9 classes. Table A9 shows that the greedy search is as competitive as the full grid search despite the time complexity reduction by several orders of magnitude and the superclass-level search is also effective to reduce cost with competitive accuracies. Note that the superclasses are identified by the feature similarity of class signatures.

**Additional Results** Table A11 presents full experimental results on the CelebA and Waterbirds datasets, which supplement Table 1 and 2. We test our robust scaling strategies (RS, IRS) with two scenarios, each aimed at maximizing worst-group or average accuracies, respectively, where each target metric is marked in blue in the tables.

# E   DISCUSSION

**Comparison to reweighting or resampling techniques** As mentioned in Section 2, most existing debiasing techniques (Sagawa et al., 2020; Liu et al., 2021; Nam et al., 2020; Seo et al., 2022a; Idrissi et al., 2022; Kirichenko et al., 2022), in principle, perform reweighting and/or resampling of training data. Our approach has a similar idea, but, instead of giving favor to the examples in minority groups during training and boosting their classification scores indirectly via iterative model updates, we directly adjust their classification scores by class-wise scaling after training, thus it gives similar but clearer effects on the results. As shown in Figure 4, although class reweighting (CR) improves the robust accuracy, this in fact identifies one of the Pareto optimal points on the trade-off curve of ERM obtained by class-specific scaling. However, because class reweighting employs a single fixed reweighting factor during training based on class frequency, it only reflects a single point and has limited flexibility compared to our wide range of scaling search. If CR employs a wide range of reweighting factors, then it can identify additional optimal points and achieve additional performance gains, but it requires training separate models for each factor, which is not realistic. Note that our

Table A11: Results of our robust scaling methods on top of various baselines on the Waterbirds dataset, which supplement Table 2. Blue color denotes the target metric that the robust scaling aims to maximize. Compared to RS, IRS improves the overall trade-off.

| Method | Robust Coverage | | Accuracy (%) | | | Accuracy (%) | | |
|---|---|---|---|---|---|---|---|---|
| | Worst-group | Unbiased | Worst-group | Unbiased | Average | Worst-group | Unbiased | Average |
| ERM | - | - | 76.3 (0.8) | 89.4 (0.6) | **97.2 (0.2)** | 76.3 (0.8) | 89.4 (0.6) | 97.2 (0.2) |
| ERM + RS | 76.1 (0.4) | 82.6 (0.3) | 81.6 (1.9) | 89.8 (0.5) | **97.2 (0.2)** | **79.1 (2.7)** | **89.7 (0.6)** | 97.5 (0.1) |
| ERM + IRS | **83.4 (1.1)** | **86.9 (0.4)** | **89.3 (0.5)** | **92.7 (0.4)** | 94.1 (0.3) | 77.6 (7.0) | 89.6 (1.1) | **97.5 (0.3)** |
| CR | - | - | 76.1 (0.7) | 89.1 (0.7) | **97.1 (0.5)** | 76.1 (0.7) | 89.1 (0.7) | 97.1 (0.3) |
| CR + RS | 73.6 (2.3) | 82.0 (1.5) | 79.4 (2.4) | 89.4 (1.0) | 96.8(0.8) | 76.4(1.5) | **89.3 (0.8)** | 97.5 (0.3) |
| CR + IRS | **84.2 (2.5)** | **88.3 (1.0)** | **88.2 (2.7)** | **92. 1(0.7)** | 95.7 (1.1) | 77.3 (4.7) | 88.6( 1.2) | 97.4 (0.2) |
| SUBY | - | - | 72.8 (4.1) | 84.9 (0.4) | 93.8 (1.5) | 72.8 (4.1) | 84.9 (0.4) | 93.8(1.5) |
| SUBY + RS | 72.5 (1.0) | 81.2 (1.4) | 75.9 (4.4) | 86.3 (0.9) | **95.2 (1.4)** | 70.7 (5.8) | 85.4 (1.6) | 95.5 (0.2) |
| SUBY + IRS | **78.8 (2.7)** | **85.9 (1.0)** | **82.1 (4.0)** | **89.1 (0.9)** | 92.6 (2.2) | **74.1 (4.1)** | **86.3 (0.9)** | **96.2 (0.6)** |
| SUBG | - | - | 86.5 (0.9) | 88.2 (1.2) | 87.3 (1.1) | **86.5 (0.9)** | **88.2 (1.2)** | 87.3 (1.1) |
| SUBG + RS | 80.6 (2.0) | 82.3 (2.0) | 87.1 (0.7) | **88.5 (1.2)** | 87.9 (1.1) | 74.0 (5.6) | 85.9 (2.8) | 91.3 (0.4) |
| SUBG + IRS | **82.2 (0.8)** | **84.1 (0.8)** | **87.3 (1.3)** | 88.2 (1.2) | 87.6 (1.2) | 70.2 (1.6) | 84.5 (1.0) | **93.5 (0.4)** |
| GR | - | - | 86.1 (1.3) | 89.3 (0.9) | **95.1 (1.3)** | **86.1 (1.3)** | 89.3 (0.9) | 95.1 (1.3) |
| GR + RS | 83.7 (0.3) | 86.8 (0.7) | **89.3 (1.3)** | 92.0 (0.7) | 93.1 (3.2) | 82.2 (1.3) | **90.8 (0.5)** | 95.4 (1.3) |
| GR + IRS | **84.8 (1.7)** | **87.4 (0.4)** | 89.1 (0.8) | **92.2 (1.0)** | 92.9 (2.1) | 82.1 (1.4) | 90.5 (0.7) | **95.6 (0.8)** |
| GroupDRO | - | - | 88.0 (1.0) | 92.5 (0.9) | 95.8 (1.8) | **88.0 (1.0)** | **92.5 (0.9)** | 95.8 (1.8) |
| GroupDRO + RS | 83.4 (1.1) | 87.4 (1.4) | 89.1 (1.7) | 92.7 (0.8) | **96.4 (1.5)** | 80.9 (4.4) | 91.3 (1.0) | **97.1 (0.3)** |
| GroupDRO + IRS | **86.3 (2.3)** | **90.1 (2.6)** | **90.8 (1.3)** | **93.9 (0.2)** | 96.0 (0.6) | 83.2 (1.7) | 91.5 (0.8) | **97.1 (0.4)** |

method can be easily applied to CR or other methods, which allows us to identify more desirable optimal points on the trade-off curve with negligible computational overhead.

**Limitation**   Although our framework is simple yet effective for improving target metrics with no extra training, it does not learn debiased representations as it is a post-processing method. However, this suggests that existing training approaches may also not actually learn debiased representations, but rather focus on prediction adjustment for group robustness in terms of robust accuracy. To address this concern, we introduce a comprehensive measurement that enables a more accurate and fairer evaluation of base algorithms, which considers the full landscape of trade-off curve. In Appendix B., we noted the limitation of our attribute-specific robust scaling.

