# OpenReview forum: "Group Robustness via Adaptive Class-Specific Scaling"
_ICLR.cc/2024/Conference — ICLR 2024 Conference Withdrawn Submission_

### Official Review · Reviewer_mojf · 2023-10-25

**Soundness:** 2 fair
**Presentation:** 2 fair
**Contribution:** 2 fair
**Rating:** 5
**Confidence:** 4

**Summary:**

This paper proposes a simple class-based post-hoc logit scaling strategy to optimize the trade-off between average and robust accuracy using held-out group-annotated data. The robust coverage metric is proposed to quantify this trade-off, and the methods achieve consistent performance gains compared to recent algorithms on 4 different datasets.

**Strengths:**

1. This paper deals with an important problem in group robustness, which is the trade-off between robust and average accuracies: while literature has generally focused on maximizing robust accuracy only, practice demands balancing the two objectives, and this paper fills the gap.
2. The proposed robust scaling method is post-hoc and training-free, which means computational cost is negligible and it can easily be added to existing training pipelines. Also, the robust coverage metric is intuitive and seems useful for evaluation of future algorithms in this area.
3. Benchmark evaluation is comprehensive, covering 4 well-known datasets and recent competitive methods, and using means and standard deviations over 3 seeds. The proposed methods achieve consistent (though often modest) performance gains, and thoughtful hyperparameter tuning analysis is provided.

**Weaknesses:**

1. The simplicity of the algorithms is a strength of the paper, but for such simple algorithms I would expect detailed ablations and analysis beyond benchmarks and hyperparameter tuning. In particular, I have some concerns about how the scaling vector is found and what kinds of characteristics it displays, which calls into doubt both the novelty and soundness of the contribution.

    * The actual algorithm for finding the scaling vector is not really described - Section 3.2 just says that there is a greedy search where the best scaling factor is identified one class at a time. How exactly is this search performed? How is the best scaling factor identified? How does one select the order in which the classes are optimized (and does this matter)? Precisely how is this algorithm extended to the clusters in IRS?

    * The actual reason for the good performance of robust scaling is insufficiently explored. What kinds of solutions do RS and IRS find, and how do they compare to the proportion of classes in the training/validation set? What if one just uses the inverse of the class proportions in the training/validation set as the class scaling vector (is it similar to the CR solution)? How does class scaling correspond to increased worst-group accuracy, for example, does RS upweight the worst group more than CR does? Is there a pattern to the different class scaling vectors found at different points on the Pareto frontier? Finally, on page 5 the authors justify the greedy search by saying that there are many different near-optimal solutions: what does it mean to have many different near-optimal solutions, and why does this justify a greedy algorithm (certainly there are problems for which there exist many near-optimal solutions but a greedy algorithm does not find them)?
2. The ERM and CR comparisons in the tables are trained on only the training set, as far as I understand, but RS and IRS utilize additional data (the validation set) to find the best class scaling. While RS and IRS can be implemented on top of an already-trained model, so this comparison makes sense, I believe it is also necessary to have an additional comparison to ERM and CR trained on the training set plus this additional held-out data. In essence, this would answer the question: should held-out data be used for RS/IRS, or should one just continue training the ERM model for some additional epochs with the held-out data included?
3. For CivilComments and FMOW, it would be helpful to have at least one additional comparison to relevant methods, since the current comparisons only list GR and GDRO. For example, [8] may be a relevant comparison for CivilComments.

**Questions:**

1. There are a few minor typos: extra comma on Section 3.1 line 4; missing “the” in Section C.2 line 2; unneeded hyphen in “robust-scaling” on page 1 line 3 from bottom; should be “trade-offs of existing algorithms” on page 1 line 10-11 from bottom; use “assign” instead of “are assigning” and “sample” instead of “samples” in Section 2 lines 7-8; should be “DistilBert” instead of “DistillBert” in Section 4.1 line 4; should be “$q=0.5$” and “$\eta=0.1$” instead of “0.5 of $q$” and “0.1 of $\eta$” in Section C.3.
2. The sentence starting with “It allows us to identify” on page 1 is confusing and could use a reword.
3. The authors may consider citing [1] or similar in page 1 paragraph 1 for ERM and [2] in page 1 paragraph 2 for evidence that spurious correlation leads to poor generalization in minority groups.
4. For Section 3.1 line 5, note that some group robustness algorithms don’t require any group annotations for model selection, e.g., MaskTune from [3], DivDis from [4], or class-balanced last-layer retraining from [5].
5. In Equation 3, it should be written explicitly that $c \in C$.
6. In the line after Equation 3, what are prediction scores? I’m assuming they are the logits because they are in $\mathbb{R}^C$, but it also sounds like they could be the post-softmax outputs. This should be written explicitly.
7. I would suggest switching paragraphs 2 and 3 in Section 3.2. It is difficult to interpret Figure 3 without first knowing how the scaling vector is found.
8. In Equation 4, is $s$ the scaling vector? It should be explicitly denoted that $s\in \mathbb{R}^C$ under the max.
9. Section C.1 could be better organized, perhaps into a table or at least a bulleted list.
10. What data augmentation was used during training? This information should be included in Section C.
11. The bibtex could use an update; some of the papers listed as arxiv versions were accepted at conferences in 2022-23. Two arbitrary examples are [6], which was accepted at NeurIPS 2022, and [7], which was accepted at ICLR 2023.
12. It would be helpful to denote where the RS and IRS solutions fall on the Pareto curves of Figure 3 and Figure 4.

**Recommendation**

Overall, while this paper proposes a simple method that deals with an important problem in the literature, my concerns with respect to the novelty and soundness of the contribution cause me to lean slightly towards rejection rather than acceptance.

**References**

[1] Vladimir Vapnik. Statistical learning theory. Wiley, 1998.

[2] Geirhos et al. Shortcut learning in deep neural networks. Nature Machine Intelligence, 2020.

[3] Taghanaki et al. MaskTune: Mitigating Spurious Correlations by Forcing to Explore. NeurIPS, 2022.

[4] Lee et al. Diversify and Disambiguate: Learning From Underspecified Data. ICLR, 2023.

[5] LaBonte et al. Towards Last-layer Retraining for Group Robustness with Fewer Annotations. NeurIPS, 2023.

[6] Kim et al. Learning Debiased Classifier with Biased Committee. NeurIPS, 2022.

[7] Kirichenko et al. Last Layer Re-Training is Sufficient for Robustness to Spurious Correlations. ICLR, 2023.

[8] Liu et al. Just Train Twice: Improving Group Robustness without Training
Group Information. ICML, 2021.

---

### Official Review · Reviewer_cWjo · 2023-10-30

**Soundness:** 3 good
**Presentation:** 2 fair
**Contribution:** 2 fair
**Rating:** 5
**Confidence:** 4

**Summary:**

This paper introduces a post-training strategy that can improve worst-group and average-case accuracies in particularly. In addition, it provides a new unified metric that encapsulates the trade-off between two types of accuracies. The paper reveals that the proposed method outperforms several existing baselines in some datasets.

**Strengths:**

1. The proposed methods (RS and IRS) appear to be simple, yet quite effective.
2. The newly proposed metric is intriguing and effective.

**Weaknesses:**

1. This method might lack rigorous reasoning and a theoretical foundation.
2. As for the proposed evaluation metric, changes in the hyperparameter D may have an unknown impact on its effectiveness.
3. Some key details of the method, including how to perform greedy algorithms, are not presented in a symbolic manner.
4. Evaluating the proposed method on more datasets would help strengthen the analysis, especially on those datasets with imbalanced samples.

**Questions:**

1. Could the authors provide more details about how the greedy algorithm works?
2. K-means clustering algorithm has certain limitations. In IRS method, have you considered trying other clustering algorithms? What would be the impact on the experimental results with different clustering algorithms?

---

### Official Review · Reviewer_SN9B · 2023-10-31

**Soundness:** 3 good
**Presentation:** 3 good
**Contribution:** 3 good
**Rating:** 6
**Confidence:** 3

**Summary:**

The paper suggested a naïve ERM baseline matches or even outperforms the recent debiasing methods by only adopting the class-specific scaling technique. The authors extended global class-wise scaling into instance-wise scaling by running clustering and then finding the optimal scaling for each cluster. Finally they introduced a unified metric that summarizes the trade-off between the two accuracies as a scalar value.

**Strengths:**

* The proposed method is simple and easy to implement. The instance-wise version does not add too much computational cost as it scales linearly w.r.t. the number of clusterings.

* Despite of being straightforward, the performance of the proposed method is good.

* The experiments are comprehensive.

**Weaknesses:**

* Is $\hat{y}$ in Eq. 3 pre- or post- the softmax operator? if it comes after softmax, shouldn't there be a non-negative constraint over $s$?

* "To identify the optimal scaling factor $s$, we perform a greedy search, where we first identify the best scaling factor for a class and then determine the optimal factors of the remaining ones sequentially conditioned on the previously estimated scaling factors. " does it mean that the optimised $s$ somehow depends on the *order* of classes that the search works on.

* As K-means itself is a non-deterministic algorithm, the authors need to investigate how robust the instance-wise scaling w.r.t. random init. of K-means.

**Questions:**

Please see above.

---

### Official Review · Reviewer_SBij · 2023-10-31

**Soundness:** 1 poor
**Presentation:** 1 poor
**Contribution:** 1 poor
**Rating:** 3
**Confidence:** 4

**Summary:**

The paper aims to improve model robustness against spurious correlations. It argues that existing approaches improve the robust accuracy (e.g., worst-group accuracy) at the expense of average accuracy. To address this, it uses the validation set (with domain/group label) to greedy-search a classwise scaling factor for each metric, such that the metric calculated based on the scaled prediction is maximized in the validation set. Then when testing on each metric, it uses the corresponding scaling factor. Combining the proposed method with existing baselines leads to consistent improvements in standard benchmarks.

**Strengths:**

The paper tackles an important problem. It has a detailed discussion of related works. Performance improvements are consistent.

**Weaknesses:**

1. The proposed method is too trivial. It simply searches a classwise scaling factor on the validation set. There is also no insight provided, e.g., why does the proposed classwise scaling bring improvements in the target metric? How does the optimal scaling factor differ for various metrics?

2. The comparison with baselines is potentially unfair. Since the scaling factors are found by greedy search on the validation set, they can be regarded as trainable parameters. Hence the proposed method effectively leverages the validation set as additional training data. Hence direct comparison with baselines is unfair.

3. The paper is poorly written. For example, in the first paragraph of the introduction, the concept of "group" and "minority group" is discussed without proper introduction. The claim that "there has been little exploration for a comprehensive evaluation that considers both metrics jointly" is not convincing, as there are many works studying the tradeoff between in-distribution and out-of-distribution accuracy [1*]. Despite the simplicity of the method, it is poorly explained, e.g., what is the detailed procedure of greedy search? Is it by gradient descent or brute-force trial from a pre-determined list? Section 3.3 is especially confusing: what are the cluster centroids used for? How to perform step 2? How to determine the cluster membership?

[1*] Calibrated ensembles can mitigate accuracy tradeoffs under distribution shift. UAI 2022.

Other suggestions & questions

1. It is recommended to include the dataset statistics (train/val/test set), especially because the proposed method leverages the val set.

2. Beyond numerical results, it is more interesting to see some qualitative examples of how the proposed approach improves baselines.

**Questions:**

See weaknesses.